# Carbohydrates and Endurance Exercise: A Narrative Review of a Food First Approach

**DOI:** 10.3390/nu15061367

**Published:** 2023-03-11

**Authors:** Alireza Naderi, Nathan Gobbi, Ajmol Ali, Erfan Berjisian, Amin Hamidvand, Scott C. Forbes, Majid S. Koozehchian, Raci Karayigit, Bryan Saunders

**Affiliations:** 1Department of Exercise Physiology, Borujerd Branch, Islamic Azad University, Borujerd 6915136111, Iran; 2Applied Physiology and Nutrition Research Group, School of Physical Education and Sport, Rheumatology Division, Faculdade de Medicina FMUSP, Universidade de São Paulo, São Paulo 01246-903, SP, Brazil; 3School of Sport, Exercise and Nutrition, Massey University, Auckland 0745, New Zealand; 4Department of Exercise Physiology, Faculty of Physical Education and Sport Sciences, University of Tehran, Tehran 1415563117, Iran; 5Department of Biological Sciences in Sport, Faculty of Sport Sciences and Health, Shahid Beheshti University, Tehran 1983969411, Iran; 6Department of Physical Education Studies, Faculty of Education, Brandon University, Brandon, MB R7A6A9, Canada; 7Department of Kinesiology, Jacksonville State University, Jacksonville, AL 36265, USA; 8Faculty of Sport Sciences, Ankara University, Gölbaşı, Ankara 06830, Turkey; 9Institute of Orthopaedics and Traumatology, Faculty of Medicine FMUSP, University of São Paulo, São Paulo 01246-903, SP, Brazil

**Keywords:** carbohydrates, exercise performance, sport foods, endurance athletes, cycling, running

## Abstract

Carbohydrate (CHO) supplements such as bars, gels, drinks and powders have become ubiquitous as effective evidence-based CHO sources that improve endurance exercise performance. However, athletes are increasingly turning to more cost-effective ‘food-first’ approaches for CHO ingestion to improve exercise performance. Mixed CHO foods including cooked lentils, oats, honey, raisins, rice, and potatoes are all effective pre-exercise CHO food sources. Caution is advised when selecting some of these foods as a primary CHO source, as some athletes may be prone to gastrointestinal discomfort—especially regarding those foods where the quantities required for recommended CHO intake may be voluminous (e.g., potatoes). Palatability may be another barrier to the ingestion of some of these CHO-rich foods. Although most of these CHO-rich foods appear effective for exercise performance or recovery when consumed pre- and post-exercise, not all are viable to ingest during exercise due to difficulties in the quantities required, transport, and/or gastrointestinal discomfort. Raisins, bananas and honey may be particularly useful CHO foods for consumption during exercise, as they are easily transportable. Athletes should trial CHO food sources before, during and/or following training before implementation during competition.

## 1. Introduction

Carbohydrate (CHO) provision for exercise performance has become a requisite for competitive athletes, with the amount of CHO required intrinsically linked to the intensity and duration of exercise [1]. CHO may be consumed pre-exercise, during exercise, and post-exercise throughout training and competition, each of which will have implications as to the efficacy of the physiological responses and adaptations. There is some suggestion that periodised CHO ingestion may be beneficial for endurance athletes throughout training [2], though evidence to support this theory currently remains limited. Pre-exercise CHO ingestion can begin in the days leading into the exercise event to ensure that muscle glycogen stores are maximized [3,4]. Additionally, CHO ingestion provided up to 3–4 h prior to exercise is likely to increase muscle glycogen content. During exercise, CHO intake maintains blood glucose and/or provides fuel for oxidation, thus sparing muscle and liver glycogen [5,6]. Finally, CHO taken post-exercise aims to replenish both muscle and liver glycogen stores [7]. The speed of recovery depends on the timing and quantity of CHO [8], such that rapid and large quantities of CHO may be required to optimise performance during a subsequent exercise bout performed within hours of the previous exercise bout [9]. Thus, CHO ingestion is an important nutritional aspect to consider for both training and competition.

CHO supplements have become commonplace among athletic populations, with numerous commercially available products, including cereal bars, gels, drinks and powders, considered effective evidence-based CHO sources to improve endurance exercise performance [10,11]. Despite the practicalities of employing such CHO-rich products, particularly during exercise, a ‘food-first’ approach to CHO ingestion for exercise may be of great relevance since acquiring CHO via dietary sources will also lead to co-ingestion of other important macro- (e.g., proteins and lipids) and micro- (e.g., vitamins and minerals) nutrients which are of benefit to athletes [12,13,14]. Further, athletes may wish to prioritise food over supplements for a multitude of personal reasons, including food choices (e.g., animal or plant-based), taste, gastrointestinal discomfort, cost, sustainability, behaviour, health and religion [12,15,16,17,18]. Dietary sources of CHO are numerous, including lentils, bananas, oats, honey, raisins, potatoes, rice, pasta. Each of these foods has its unique macronutrient and micronutrient content that will modify the speed at which they increase glucose in the bloodstream (i.e., their glycaemic index [GI]). The GI is a rating system based upon how much blood glucose is increased by ingesting specific foods, categorized into low-to-moderate and high GI foods. Foods with different GI values lead to a different metabolic response [19]; however, the GI of a pre-exercise meal has no clear benefit for endurance performance [20]. In contrast, high-GI CHO foods ingested during recovery between exercise bouts may accelerate post-exercise muscle glycogen storage [21] and improve subsequent exercise performance [22]. Thus, different CHO-food sources with different GI may be more or less efficient for exercise performance and glycogen replenishment when recovery time is short. However, the feasibility of implementing each dietary CHO source for pre-exercise, during exercise, and for post-exercise consumption is an important consideration for athletes. This narrative review aimed to determine whether a food-first approach to CHO provision for endurance exercise is an appropriate means to fuel endurance exercise and optimise performance compared to traditional CHO supplements.

## 2. Pre-Exercise CHO Ingestion

Initial studies tested different pre-exercise GI foods on substrate utilization and exercise performance mediated by different insulin responses, with equivocal findings between low- and high-GI foods [23,24]. While some evidence showed positive effects after consuming low-GI foods, other studies reported no difference between pre-exercise low-GI foods compared to high-GI foods [20,25]. The consumption of 1–4 g·kg^−1^ body mass (BM) of CHO is generally recommended 1 to 4 h prior to endurance exercise (Figure 1) [26]. Pre-exercise food selection depends on various factors such as sex, training status, and/or habitual dietary intake of endurance athletes [27]; however, regardless of individuals’ choices, pre-exercise ingestion of CHO appears to be important. For example, combining a 2.5 g CHO·kg^−1^ BM meal before exercise with a CHO drink (6.9% CHO) during exercise was better for performance compared to CHO ingestion during exercise alone, or no CHO (placebo) during a run to exhaustion at 70%V̇Omax [28] (Table 1). Aandahl et al. [29] compared a high (3 g·kg BM) and low (0.5 g·kg^−1^ BM) CHO pre-exercise meal ingested ~3.5 h before exercise on physiological variables and time-to-exhaustion during a graded exercise test. Recreational and well-trained endurance athletes were recruited and also performed a trial while fasting. The high-CHO meal improved exercise performance relative to the low-CHO meal and the fasting state, although no differences in physiological responses were shown. These performance effects were evident for both trained and recreational athletes, demonstrating that a high CHO pre-exercise meal appears to be better for performance than a low-CHO meal or nothing.

When considering pre-exercise CHO intake, many individuals will indicate foods such as potatoes, rice, and pasta as “typical” CHO sources. Potatoes are predominantly composed of water with only ~20 g of CHO per 100 g of boiled potato. Both white and brown rice consist predominantly of water (~69–70%) and contain an average of 28 g and 26 g of CHO per 100 g. White or “refined” pasta is composed of ~67% water, and 100 g of plain cooked spaghetti provides approximately 26 g of CHO. Cooked lentils are another CHO food [30] and are composed of ~19.5% CHO. All these are viable options as pre-exercise CHO meal options for athletes aiming to optimise their exercise performance, although athletes should be aware that they are relatively low in CHO per total volume, with only 20–30% comprised of CHO (Figure 2).

Thomas et al. [31] recruited eight trained cyclists who pedalled to exhaustion at 65–70% V̇O_2max_ following the ingestion of either lentils, potatoes, glucose or water only, provided 1 h before exercise. Each meal provided 1 g·kg^−1^ BM of CHO. The volunteers ingested approximately 70 g of CHO, meaning they had to consume ~650 g of cooked potatoes and ~430 g of cooked lentils. Plasma glucose peaked ~45 min after ingestion with potatoes, likely due to its high-GI leading to the rapid absorption of CHO. The plasma glucose and insulin responses were lower following lentils ingestion, and CHO oxidation was lower during exercise with lentils compared to the glucose and potato conditions. Importantly, time-to-exhaustion was greater in the lentil condition compared to the other CHO conditions and water, while the glucose and potato conditions did not significantly improve performance compared to water. These findings suggests that lentils may be an effective pre-exercise CHO food source to be ingested alone, or as part of a mixed-CHO meal, if ingested 1–3 h before exercise, and may enhance metabolic responses (higher fat oxidation, lower insulin, and CHO oxidation) [32] during submaximal exercise compared to other CHO food sources.

Bananas are a CHO-rich fruit containing a mixture of glucose, fructose and sucrose. Mitchell et al. [33] compared various sources of pre-exercise (−60 min) CHO ingestion in trained runners in a hot environment (32 °C, 65% relative humidity). The CHO conditions included banana slurries (banana and water; 54 g CHO), a CHO solution mixture of glucose and fructose (54 g CHO), a high fructose corn syrup solution (72 g CHO), a glucose-only solution (54 g CHO), a saccharose and glucose mixture solution (54 g CHO), and a placebo drink (water identical in flavour, texture and colour). The different types of CHO altered the blood glucose response compared to the placebo drink, although this had no influence on 10 km running performance with no differences between any condition. A higher fluid retention was shown with the glucose/fructose and glucose-only solutions, likely due to the high sodium content [33]. The lack of a performance improvement between CHO forms compared to placebo is perhaps unsurprising, since CHO ingestion prior to short-duration exercise (<1 h) may not be necessary [1], and makes it difficult to speculate as to the efficacy of pre-exercise CHO from bananas.

Raisins are a type of sun-dried grape and a rich source of CHO [34]. Kern et al. [35] compared the pre-exercise ingestion of raisins to a CHO sport gel on 45 min cycling at 70% V̇O_2max_, followed by a 15 min performance time-trial (TT). Eight trained endurance cyclists ingested 1 g·kg^−1^ BM CHO from either raisins or sports gel 45 min prior to the test and showed a similar amount of total work carried out between the raisin and sports gel, meaning that pre-exercise CHO provision via raisins and sports gels produced an equal power output during the exercise. Thus, raisins appear to be an appropriate alternative pre-exercise CHO food source to CHO supplements, though more confirmatory studies are warranted.

Oats are a rich source of CHO, providing ~68 g CHO per 100 g [36]. Paul et al. [37] examined the effects of three isoenergetic pre-exercise meals consisting of oats (containing 41.5 g CHO), wheat (containing 53.9 g CHO) and corn cereals (containing 54.7 g CHO) plus skimmed milk, compared to a fasting control trial. Twelve healthy adults ingested the pre-exercise meals 90 min before 90 min of steady-state cycling at 60% V̇O_2peak_ followed by a 6.4 km TT. There was no performance improvement with any meal compared to the fasted trial, though it must be acknowledged that there was ~12 g less CHO in the oat meal versus the other meals. Despite no performance effect, the feeling of fatigue was greater in the fasted trial, measured using the profile of mood states (POMS) questionnaire, than in the other treatments. Jones et al. [38] compared three available “cost-effective” oat- and wheat-based CHO meals relative to a commercial sports bar. Eight endurance-trained males ingested four isoenergetic meals including (i) a semi-liquid oat-based combination (77% CHO), (ii) a semi-liquid oat-based CHO (68% CHO), (iii) a semi-liquid wheat-based CHO) (75% CHO), and (iv) a dense solid, fructose-based sports bar (69% CHO). Meals were provided 2 h before a 60 min self-paced cycle ergometer test. Regardless of which meal was ingested, there was no difference in heart rate, oxygen consumption, respiratory exchange ratio or exercise performance. These data suggest that each CHO source was equally beneficial for this type of exercise, although this study is limited by the lack of a control treatment to determine whether CHO is beneficial compared to no CHO. Kirwan et al. [39] investigated the effect of oat-based breakfast meals provided 45 min prior to a cycle to exhaustion at 60% V̇O_2peak_ in recreationally active women. The breakfasts consisted of sweetened whole-grain rolled oats (75 g CHO + 7 g fibre), sweetened whole-oat flour (75 g CHO + 3 g fibre) or 300 mL of water as a control. Exercise performance improved by 16% and 10% in the rolled and flour oat conditions compared to control, with no difference between meals. In a similar study, 75 g CHO in regular whole grain rolled oats mixed with 300 mL of water improved cycling exercise to exhaustion at 60% V̇O_2peak_ compared to water alone (control) when ingested 45 min before exercise [40]. Another study by the same research group showed that 75 g of CHO ingested as rolled oats improved cycling time-to-exhaustion at 60% V̇O_2peak_ compared to control, while 75 g in puffed rice did not. The data from these studies suggest that oats may be a suitable pre-exercise CHO source for endurance exercise. Importantly, oats are often consumed mixed with other foods such as honey, banana, milk, cherries and chia seeds to enhance palatability. Work is needed to consider these kinds of mixed meals on metabolism and exercise performance.

One study investigated the effect of a pre-exercise meal containing rice on 21 km running performance in eight endurance-trained male runners [41]. Meals were provided 2 h prior to exercise and consisted of a non-CHO low energy jelly (control), or one of two dishes equivalent in calories and CHO content (61%), namely a high-GI meal containing 92 g of jasmine rice and a low-GI meal containing no rice. A CHO-electrolyte drink (6.6% glucose) was also provided every 2.5 km throughout the exercise at 2 mL·kg^−1^ BM in all trials (total ~74 g CHO). Time-to-complete the 21 km run was not different between the CHO meals, although only the rice-based meal improved performance compared to control. These data provide some evidence to suggest that a pre-exercise CHO meal containing rice may be a useful strategy for endurance exercise performance. However, rice contains a substantial amount of water, meaning that to obtain CHO intakes of 30 g it would be necessary to ingest approximately 106 g of white rice or 116 g of brown rice (Figure 2). This may trigger some gastric discomfort due to the amount of food needed to achieve the desired quantity of CHO, and individuals are encouraged to determine their individual response to consumption of this food.

Traditional CHO food sources such as pasta, lentils, potato and rice may be interesting CHO sources to implement pre-exercise, although individuals should be wary of the amount of CHO in some of these as they are relatively low compared to other sources (Figure 2). This means that the consumption of large quantities of food may be necessary to achieve the desired CHO levels (e.g., ~150 g of potatoes per 30 g of CHO; Figure 2), but this could lead to gastric discomfort [42]. More studies are needed to determine whether a single CHO source like pasta, lentils, rice, or potato alone is a more viable pre-exercise CHO loading strategy versus mixed high CHO meals. Bananas may also be considered as a pre-exercise CHO snack to be ingested <2 h before exercise for those athletes that feel hungry close to exercise. Furthermore, those wishing to add supplemental CHO to foods to achieve the desired CHO dose may do so [43].

**Table 1 nutrients-15-01367-t001:** Summary of studies exploring the effects of pre-exercise CHO food ingestion on endurance exercise performance.

Study	Subjects	Exercise Test	Treatments	Timing	Performance
Thomas et al. [31]	8 trained male cyclists (V̇O_2max_: 62.5 ± 3.7 mL·kg^−1^·min^−1^)	Cycling to exhaustion at 65–70% V̇O_2max_	L: 1 g·kg^−1^ CHO from lentils (LGI) P: 1 g·kg^−1^ CHO potato (HGI) G:1 g·kg^−1^ CHO glucose W: Water	1 h before exercise	↑L vs. G ↑L vs. P ↑L vs. W
Paul et al. [37]	12 healthy adults (six women V̇O_2peak_: 50.6 ± 4.3; six men V̇O_2peak_:58.3 ± 5.1 mL·kg^−1^·min^−1^	90 min of steady state cycling at 60% V̇O_2peak_ followed by a 6.4 km TT	Oats (O): 41.5 g CHO, 5.7 g fibre, 3.8 g fat, 10 g protein Wheat (W): 53.9 g CHO, 7.2 g fibre, 1.4 g fat, 7.2 g protein Corn (C): 54.7 g CHO, 0.7 g fibre, 0 g fat, 4.5 g protein Fasted (F)	90 min before cycling	↔O vs. W vs. C
Chryssanthopoulos et al. [28]	10 male recreational runners (V̇O_2max_: 58.6 ± 1.9 mL·kg^−1^·min^−1^)	Treadmill running at 70% V̇O_2max_ to exhaustion	M + C: Pre-exercise 2.5 g·kg^−1^ BM CHO [white bread, jam, cornflakes, sugar, skimmed milk, orange juice], and CHO drinks during exercise [6.9% CHO: dextrose, maltodextrin, and glucose syrup] P + C: Pre-exercise PL and CHO ingestion during exercise P + P: Pre-exercise PL and PL drink during exercise	3 h before exercise and during exercise	↑M + C vs. P + C ↑M + C vs. P + P ↑P + C vs. P + P
Jones et al. [38]	8 endurance trained males	60 min self-paced cycle	(Combo): Semi-liquid oat-based CHO/fat/protein combination (77% CHO, 7% protein and 5% fat) [Oatmeal, sugar, whole wheat, brown sugar, dried bananas, barley flakes, wheat farina, almonds, and guar gum] (O): Semi-liquid, oat-based CHO (68% CHO,14% protein and 1.7% fat) [Whole-grained rolled oats, calcium carbonate, and guar gum] (W): Semi-liquid wheat-based CHO (75% CHO, 11% protein) [Wheat farina, wheat germ, salt, guar gum (460 kcal] (Bar): Dense solid, fructose-based CHO/protein/vitamin (69% CHO, 14% protein, 3% fat) [High fructose corn syrup, fruit juice concentrate, oat bran, malto-dextrins, milk protein, banana, cashew butter, rice]	2 h before the test	↔Combo vs. O vs. W vs. Bar
Kirwan et al. [39]	6 recreationally active women (V̇O_2max_: 48.3 ± 3.0 mL·kg^−1^·min^−1^)	TTE at 60% V̇O2max	(R): 75 g CHO [sweetened whole-grain rolled oats with 7 g fibre + 300 mL water] (F): 75 g CHO [sweetened whole-oat flour with 3 g fibre + 300 mL water] (C): Control [Water]	45 min before exercise	↑R vs. C ↑F vs. C
Mitchell et al. [33]	10 trained runners (V̇O_2max_: 59.35 ± 2.46 mL·kg^−1^·min^−1^)	10 km treadmill run	HGF: 72 g CHO high fructose corn syrup solution LFG: 54 g CHO low volume glucose/fructose GLU: 54 g CHO glucose solution SUG: 54 g CHO sucrose/glucose mixture BAN: 54 g CHO banana with water (900 mL) WP: Water PL	1 h before exercise	↔HGF vs. LFG vs. GLU vs. SUG vs. BAN vs. WP
Kirwan et al. [40]	6 males (V̇O_2peak_: 54.3 ± 1.2 mL·kg^−1^·min^−1^)	Cycling to exhaustion at 60% V̇O_2peak_	(O): 75 g CHO [rolled oats as a moderate GI meal (~61) + 300 mL water] (P): 75 g CHO [Puffed rice as a HGI meal (~82) + 300 mL of water] (C): Control [Water]	45 min before exercise	↑O vs. C ↔P vs. C
Chryssanthopoulos et al. [44]	10 male recreational runners (V̇O_2max_: 63.5 ± 2.3 mL·kg^−1^·min^−1^)	Treadmill running at 70% V̇O_2max_ to exhaustion	M + C: Pre-exercise 2.5 g·kg^−1^ BM CHO [white bread, jam, cornflakes, sugar, skimmed milk, orange juice], and CHO drinks during exercise [6.9% CHO: dextrose, maltodextrin, and glucose syrup] M + W: Pre-exercise 2.5 g·kg^−1^ BM CHO [white bread, jam, cornflakes, sugar, skimmed milk, orange juice], and water during exercise P + W: Pre-exercise PL and water during exercise	3 h before exercise and during exercise	↑M + C vs. P + W ↑M + W vs. P + W
Kern et al. [35]	8 endurance-trained cyclists 4 males (V̇O_2max_: 64.1 ± 3.4 mL·kg^−1^·min^−1^) and 4 females (V̇O_2max_: 47.5 ± 10.6 mL·kg^−1^·min^−1^)	Cycling for 45 min at 70% V̇O_2max_ followed by 15 min performance trial	R: 1 g·kg^−1^ BW CHO raisin S: 1 g·kg^−1^ BW CHO sport gel	45 min before exercise	↔R vs. S
Chen et al. [41]	8 endurance-trained male runners (V̇O_2max_: 58.5 ± 1.6 mL·kg^−1^·min^−1^)	21 km performance run on a level treadmill	HGI meal: [92 g jasmine rice, 90 g parsnips, 265 g orange soda, 55 g canned lychees, 40 g ham, 35 g fish sticks, 80 g egg, 573 g water] LGI meal: [290 g clear chicken broth, 251 g soymilk, 54 g hard boiled egg, 38 g fish sticks, 46 g green peas, 81 g mung bean thread noodles, 467 g water] CON: Control [9 g low-energy sugar-free jelly]	2 h before the test meal was consumed and 2 mL·kg^−1^ BM of 6.6% CHO solution was consumed immediately before exercise and every 2.5 km afterward	↔LGI vs. HGI: ↑HGI vs. CON
Aandahl et al. [29]	11 well-trained (V̇O_2max_: 71.9 ± 5.1 mL·kg^−1^·min^−1^) and 10 recreationally trained (V̇O_2max_: 46.9 ± 2.5 mL·kg^−1^·min^−1^) men	Five submaximal 5 min constant-velocity bouts of increasing intensity and a graded exercise test to measure TTE (running on a tread-mill)	High CHO meal: 3 g·kg^−1^ BM CHO [white bread, jam, skimmed milk, oats, banana, and raisins] Low CHO meal: 0.5 g·kg^−1^ BM [yogurt, almonds, and avocado] Fasted state	3.5 h before the exercise	↑High CHO meal vs. low CHO meal ↑High CHO vs. fasted state

CHO carbohydrate, V̇O_2max_ maximal oxygen consumption, V̇O_2peak_ peak oxygen consumption, TT time-trial, TTE time to exhaustion, VA voluntary activation, CAR central activation ratio, MVC maximum voluntary contraction, sMVC sustained MVC, WE work economy, HGI high-glycaemic index, LGI low-glycaemic index.

## 3. CHO Ingestion during Exercise

In addition to pre-exercise CHO intake, CHO ingestion during endurance exercise is considered essential to maintaining performance. When a meal was provided pre-exercise (2.5 g CHO·kg^−1^ BM) and CHO given during exercise, the time-to-exhaustion during endurance exercise was 12% greater than when CHO was provided before exercise only, and 22% greater compared to a placebo [44]. When exercise lasts between 1 and 3 h, ingesting 30–60 g·h^−1^ of CHO is commonly recommended [1]. Isotonic CHO drinks containing multiple CHOs at a dose of 60–90 g·h^−1^ during exercise have been suggested to enhance endurance capacity when activity is extended above 3 h [1,45]. These larger doses of mixed CHO (e.g., glucose and fructose in a 2:1 ratio [45]) promote increased CHO absorption via two different gut transporters. Endurance athletes tend to favour transportable CHO supplements such as hydrogels, shots, bars, and chews during competition [42,46] because of higher gastrointestinal tolerability, CHO absorption and oxidation rates [47]. However, endurance athletes may see eating CHO-rich fruits and foods as a natural and cost-effective source for supplying CHO during exercise. Naturally, an important consideration is that these foods should be easily transportable for an athlete.

Bananas could be an excellent CHO source (fructose: glucose ratio of ~1:1) during exercise [48]. Nieman et al. [49] compared banana ingestion to a CHO drink on exercise metabolism and performance (Table 2). They reported no significant difference in blood glucose, blood lactate or 75 km cycling TT performance when 14 trained cyclists ingested 200 mg·kg^−1^ BM of CHO every 15 min from either bananas or a 6% CHO sports drink. Participants did report feeling significantly fuller and more bloated when consuming bananas, which may be due to the high fibre intake (~15 g). Although this did not modify exercise performance here, this may become a greater issue during longer distances requiring more CHO intake. In another study, two types of bananas (Cavendish and mini yellow) and a 6% CHO beverage led to similar performance times during a 75 km cycling TT, although none were improved compared to water alone [50]. The same research group also compared banana ingestion to pear ingestion or water alone during a 75 km cycling TT [51]. Twenty male cyclists ingested 400 mg·kg^−1^ BM CHO from either banana or pear alongside 5 mL·kg^−1^ BM of water, or water alone, 20 min prior to initiating exercise. A further 150 mg·kg^−1^ BM of CHO was ingested every 15 min throughout the test. Performance times for the banana and pear treatments were faster than water alone. These data suggest that bananas may be a good CHO source during exercise to improve endurance exercise performance. However, more research should determine the rate of CHO oxidation and exercise performance with banana ingestion compared to commonly employed CHO supplements during different types of endurance exercise tests. The size of bananas makes them easily transportable and an excellent CHO option during endurance exercise, though athletes may wish to bear in mind that they are delicate and could be damaged or undergo browning, thus making them less palatable to some individuals. Furthermore, carrying the number of bananas required to fuel prolonged exercise may not always be practical, especially for runners or during competition. Nonetheless, those engaged in endurance exercise may wish to replace a number of CHO supplements with bananas depending upon how many they can practically carry on them (Figure 3A).

Honey is a natural food made by honeybees via nectar sourced from flowers [57]. CHO is the main constituent of honey (~80–85%), including fructose, glucose, small amounts of sucrose, and varying amounts of maltose based on the botanical source of the different regions of honey breeding [58]. Two studies have shown performance benefits with honey ingestion during endurance cycling [52] and running [59]. Earnest et al. [52] showed similar performance improvements when amateur cyclists ingested 15 g of CHO from sports gels (dextrose, high GI: 100) or honey (low-GI: 35) every 16 km during a 64 km cycling TT compared to a water-only trial. Both CHO treatments allowed participants to generate more power during the last 16 km. There was a lack of blinding due to the absence of a matching placebo in colour and taste, which means modified performance may have been due to placebo effects [60]. Nevertheless, these data do show promise for the use of honey as a CHO source during exercise and may lead to improvements similar to those seen with traditional CHO supplements. Honey certainly appears more effective than no CHO. Honey may be a particularly interesting option during exercise when high CHO quantities are required due to its high CHO content (Figure 1), and glucose and fructose constitution. It is well-recognized that CHO beverages containing both glucose and fructose at doses of 60–90 g·h^−1^ may augment endurance exercise capacity compared to glucose alone. Adding fructose to glucose would increase CHO oxidation and enhance the gastric emptying rate due to a higher absorption rate via two different intestinal transporters since glucose transport is saturated at ~60 g·h^−1^ [1,61]. Honey can easily be transported, either diluted in water or in small portable plastic sachets which athletes can chew while exercising. However, honey is considered a high FODMAP food due to its high fructose content [62] and may need to be tested by athletes in training sessions to minimize the potential risk of gastrointestinal distress, while in addition it is not suitable for vegan athletes. Similar to bananas, honey could replace some, or all, CHO supplements during exercise (Figure 3B).

Raisins are a natural CHO source that can be ingested during exercise (glucose and fructose~1.1:1 ratio [63,64,65]). Rietschier et al. [53] provided 10 male endurance-trained athletes 1.1 g·kg^−1^ BM CHO from either six servings of 28 g raisin or 26 g sport jellybeans every 20 min during a 120 min intense cycle followed by a 10 km TT. Both forms of CHO maintained similar blood glucose levels during exercise, with no significant difference in TT performance. However, the lack of a placebo or control condition in either study is a limitation that does not allow us to draw any definitive conclusions on whether raisins may be more or less effective than commercial CHO sources. Too et al. [54] compared raisin ingestion to CHO-rich sport chews and water drinks. Eleven competitive endurance runners were provided 500 mg·kg^−1^ BM before exercise and 200 mg·kg^−1^ BM every 20 min throughout 80 min treadmill running at 75% V̇O_2max_ followed by a 5 km TT. There were no significant metabolic or performance differences between the raisins and sport chew trials; however, both CHO trials led to a greater performance compared to water alone. Corinthian currant [55], a dried grape derived from black grapes, has also shown similar performance improvements compared to a glucose drink (1.5 g·kg^−1^ BM) during a cycle to exhaustion at 95% V̇O_2max_ following 90 min of cycling at 60–70% V̇O_2max_. These data provide evidence that raisins can be considered an alternative to common CHO supplements during endurance exercise. They are also CHO dense (Figure 2), easily transportable as either food or drink, and are generally well-tolerated without significant gastrointestinal discomfort [54,55].

Transportability and digestibility are key when considering CHO ingestion throughout exercise. Although foods such as potatoes, rice and pasta in their own right might be appropriate CHO sources during exercise, practical limitations might not allow their implementation in practice. For example, Salvador et al. [56] compared the performance effects of CHO obtained from mashed potatoes with an equivalent of CHO from a gel. Cyclists performed 120 min of intermittent cycling at 60–85% V̇O_2peak_ before a TT to complete 6 kJ·kg^−1^ BM and ingested 15 g of CHO from potato or a CHO gel every 15 min throughout the intermittent test. To obtain 15 g of CHO within each form, participants had to ingest 128.5 g of mashed potato but only 23 g of CHO gel. TT performance was improved with CHO compared to water alone but did not differ between CHO supplement forms. Although this suggests that potato ingestion throughout the exercise was equally effective as a CHO gel, potato ingestion was associated with more gastrointestinal discomfort during the test, likely due to the greater volume ingested. Furthermore, the transport of potatoes in mashed (or any) form is far less feasible than a CHO gel, although anecdotally cyclists are known to transport baked potatoes on long rides for fuel. A similar conclusion could be reached for pasta and rice. Although rice cakes may be considered an alternative form of rice consumption, leading to similar performance times as CHO provision in gels [66], these are often made from glutinous rice powder, and it is debatable whether these can still be considered a whole food. Thus, while some data support the potential use of potatoes as a CHO source during exercise, individuals should be wary of their difficulty in transport and the quantity needed to attain suggested ingestion rates of 60–90 g·h^−1^ (Figure 1).

Collectively, these studies show that food-based sources such as banana, honey and raisins are excellent alternative CHO sources to be ingested during exercise. They are as effective as commercial-based CHO supplements such as gels and sports drinks to improve prolonged endurance performance [35,49,51,52,53,54,56,63,67]. However, the higher potential gastrointestinal distress risk related to a greater fructose ratio, fibre content and quantity to meet standard CHO doses (60–90 g·h^−1^) with CHO foods compared to CHO supplements may highlight the more effective role of CHO supplements to be ingested during exercise [63,67,68]. Moreover, it is not clear if CHO provided at the range of 90–120 g·h^−1^ from food-based sources can still improve endurance performance without any gastrointestinal discomfort. Furthermore, studies have yet to compare CHO absorption and oxidation rates of food-based CHO sources with different GI responses versus isotonic CHO drinks.

Nevertheless, athletes can ingest a combination of foods and supplements to achieve their CHO requirements during exercise (Figure 3). In fact, food sources can be fortified with CHO to provide the desired amount. Reynolds et al. [69] showed that a natural CHO source of apple puree fortified with maltodextrin to provide an equal dose of glucose to a sports drink providing 60 g·h^−1^ in a 2:1 glucose-to-fructose ratio was equally effective for ~15 min TT performance following 120 min cycling.

## 4. Post-Exercise CHO Ingestion

Following endurance exercise the replenishment of CHO is critical, particularly when recovery time between exercise sessions is limited (<4 h). CHO provision at 1–1.2 g·kg^−1^ BM is recommended to maximize muscle glycogen replenishment and storage to optimise subsequent exercise performance [7]. High-GI CHO foods have been suggested to be preferable to low GI CHO foods to optimize post-exercise glycogen replenishment [21,70], mediated by higher insulin responses [21], which may augment subsequent endurance performance [22] (Table 3). Wong et al. [22] compared high and low-GI CHO foods during 4 h recovery after 90 min constant pace running at 70% V̇O_2max_ in endurance-trained runners. Twenty min after the first test, participants ingested either a high-GI meal (GI = 77) or a low-GI meal (GI = 37), both providing 1.5 g·kg^−1^ BM CHO. The subsequent endurance running capacity was 15% greater following the high-GI versus low-GI meal. Stevenson et al. [70] compared high versus low isocaloric GI CHO foods providing 8 g·kg^−1^ BM CHO across four meals during 24 h recovery after 90 min running on a treadmill at 70% V̇O_2max_. The next day’s run to exhaustion at 70% V̇O_2max_ was longer following low-GI vs. high-GI foods; this may have been due to a higher fat oxidation during the exercise following the low-GI diet [70]. The discrepancy between these results for high and low GI meals could be explained by the different CHO loading amounts (1.5 g·kg^−1^ BM vs. 8 g·kg^−1^ BM) and recovery times (24 h vs. 4 h) within these two studies [22,70]. Longer recovery times to ingest higher CHO amounts from low-GI CHO sources may highlight the potential metabolic benefits of low-GI CHO for endurance athletes.

Various high CHO food sources are available for endurance athletes to ingest during the short-term recovery period (Figure 1). In this regard, Murdoch et al. [71] investigated the effect of banana ingestion provided in either solid or slurry form (1.1 g·kg^−1^ BM CHO) following a 90 min run and 90 min cycle on subsequent exercise capacity. Eight highly trained triathlon athletes ingested the bananas or a placebo drink in the 20 min recovery time following the initial 180 min of exercise, before performing a cycle to exhaustion at 70% V̇O_2max_. TTE was 16 (slurried form) and 18 (solid form) min greater with the bananas compared to placebo, with no difference between the CHO forms. Although intriguing, the very short recovery time between endurance tests precluded any solid conclusions regarding banana ingestion for recovery and performance in the hours following an initial exercise bout, or subsequent next-day performance. Ahmad et al. [59] showed that participants could cover more distance during a 20 min running TT following an initial run at 65% V̇O_2max_ in the heat (31 °C, 70% relative humidity) before a 2 h rehydration phase when they were provided a honey beverage (6.8% CHO content equivalent to compensate 150% of body weight loss) compared to plain water. Thus, honey ingestion might be an interesting CHO source when recovery time is short, though it is unclear if it is as effective as traditional CHO supplement sources as a recovery aid.

Flynn et al. [72] recruited male and female recreational athletes to ingest either potato-based foods or various CHO supplements (both providing 1.6 g·kg^−1^ BM CHO) following a 90 min cycle; after 4 h recovery, they performed a 20 km cycling TT. There were no differences in muscle glycogen synthesis rate or time-trial performance between the CHO sources [72]. Cramer et al. [73] also showed no difference in muscle glycogen synthesis rates and 20 km TT endurance performance between a high-CHO fast-food diet and CHO supplements ingested during the 4 h recovery period between two exercise bouts. These data suggest that the ingestion of CHO-rich foods, such as potatoes, is equally effective for the recovery of muscle glycogen and subsequent exercise performance as traditional CHO supplements (Table 3).

Chocolate milk is a popular food-based sports beverage and comprises approximately ~11–14% CHO [36]. Chocolate milk may be an appropriate CHO beverage to support post-exercise muscle glycogen re-synthesis [74], and improve subsequent endurance exercise performance [75,76]. Several studies have compared chocolate milk versus a CHO-replacement fluid and fluid replacement drink during varying recovery periods (2–18 h) after exercise, with beneficial effects on subsequent performance compared to both CHO-replacement fluid and a fluid replacement drink [75,76,77,78]. Two recent meta-analyses and systematic reviews concluded that chocolate milk can increase muscle glycogen storage as much as an isocaloric CHO drink [74], and can improve exercise time-to-exhaustion compared to a placebo beverage, with no differences compared to a CHO sport drinks [79]. Since milk contains other important macronutrients (protein and fats) and has better hydrating properties than water [80], chocolate milk may be an interesting post-exercise CHO-replenishment strategy for athletes.

Vlahoyiannis et al. [81] investigated the effects of a high and low-GI post-exercise meal on subsequent sleep quality and next-day 5 km cycling TT performance. Recreationally trained men performed a sprint interval exercise before immediately consuming either a high-GI (GI = 109) meal, consisting of jasmine rice and vegetables, or a low-GI meal (GI = 52), consisting of parboiled rice and vegetables; both meals provided 2 g·kg^−1^ BM CHO. Next-day 5 km cycling performance was not different between meals, despite increased sleep duration and sleep efficiency, and reduced sleep onset latency, following the high-GI meal. The lack of a performance effect may be due to the exercise duration (7–8 min), which may be too short to be substantially influenced by a high- or low-GI meal. This study [81] is limited by the lack of a control trial in which no dietary CHO was provided following the sprint exercise, though the finding that a high-GI diet might improve aspects of sleep is intriguing and warrants further investigation. More studies are needed to examine the effects of CHO-based foods with different GI indexes to determine the optimal scenario under which to employ high- or low-GI CHO foods for recovery.

While high-GI foods may promote muscle glycogen synthesis when a short recovery time is needed between two exercise sessions, muscle glycogen optimization over a 24 h recovery time is more dependent on the total CHO dose ingestion. More recently, eight male endurance athletes ingested 24 h high CHO meals with doses of 5, 7 and 10 g·kg^−1^ BM in a crossover study design after a 90 min glycogen depletion cycling test [82]. The results indicated that while 7–10 g·kg^−1^ BM of CHO foods managed to restore muscle glycogen to pre-exercise levels, 5 g·kg^−1^ BM of high CHO foods was insufficient to saturate muscle glycogen storage to the pre-exercise level [82]. Food-based CHO sources appear to be an effective alternative to commercial CHO supplements for post-exercise glycogen resynthesis and subsequent endurance performance improvement (Figure 1). High-GI foods may be preferable when there is a short recovery period between exercise bouts, while total CHO consumed appears more critical when longer recovery periods are possible. Further, it is important to highlight that CHO co-ingested with protein after exercise may enhance glycogen synthesis but only when the added energy of protein is ingested in addition to, not in replacement of, carbohydrates [83].

**Table 3 nutrients-15-01367-t003:** Summary of studies exploring the effects of post-exercise CHO food ingestion on performance.

Study	Subjects	Exercise Test	Treatments	Timing	Performance
Murdoch et al. [71]	8 highly trained male triathletes (run V̇O_2max_: 68.1 ± 5.4 mL·kg^−1^·min^−1^ and bike V̇O_2max_: 67.1 ± 2.6 mL·kg^−1^·min^−1^)	90 min run (R1) followed by 90 min of cycling, both at 70% V̇O_2max_. Thereafter, workloads increased by ~5% V̇O_2max_ until exhaustion (R2)	SL: Slurry of three bananas with water SO: Three solid bananas with water PL: Artificially sweetened, flavoured, and coloured drink	During 20 min rest period between two bouts	↑SL vs. PL ↑SO vs. PL
Stevenson et al. [70]	9 male recreational athletes (V̇O_2max_: 62.1 ± 2.2 mL·kg^−1^·min^−1^)	Running for 90 min at 70% V̇O_2max_ (R1), after an overnight fast, running to exhaustion at V̇O_2max_ (R2)	72% CHO, 11% fat, 17% protein) GI = 35	During 24 h recovery	↑LGI vs. HGI: during R2
Wong et al. [22]	7 endurance-trained male runners (V̇O_2max_: 61.0 ± 5.7 mL·kg^−1^·min^−1^)	Running at 70% V̇O_2max_ on a level treadmill for 90 min (R1), followed by a 4 h recovery and a further exhaustive run at the same speed (R2)	HGI: [100 g baked potato, 55 g tomato sauce, 75 g white bread, 50 g low fat, processed cheese, 50 g watermelon, 150 g 7 up] LGI: [65 g cooked macaroni, 30 g apple slices, 30 g canned chick-peas, 50 g low-fat cheese slice, 150 g fruit-flavoured yogurt, 250 mL apple juice]	20 min after R1 (4 h recovery period)	↑HGI vs. LGI
Cramer et al. [73]	Eleven recreationally active males (V̇O_2max_: 4.2 ± 0.4 mL·kg^−1^·min^−1^, 309 ± 32 Wmax)	90 min glycogen depletion ride included a 10 min warm-up at 55%Wmax followed by a series of 10 intervals (2 min at 80% Wmax followed by 4 min at 50%Wmax). After the interval series, participants completed 8 min at 60% Wmax, followed by a final 12 min at 50%Wmax 20 km cycling TT	SS: Gatorade, Kit’s Organic, Cliff Shot Bloks, Cytomax, Power Bar Recovery, and Power Bar Energy Chews (1.54 ± 0.27 g·kg^−1^ CHO, 0.24 ± 0.04 g·kg^−1^ fat, and 0.18 ± 0.03 g·kg^−1^ protein) FF: Hotcakes, Hash brown, orange juice, hamburgers, Coke, and Fries (1.54 ± 0.27 g·kg^−1^ CHO, 0.24 ± 0.04 g·kg^−1^ fat, and 0.18 ± 0.03 g·kg^−1^ protein)	0 and 2 h post-exercise (total 4 h recovery)	↔SS vs. FF 20 km TT
Ahmad et al. [59]	10 male recreational runners (V̇O_2max_: 51.7 ± 4.1 mL·kg^−1^·min^−1^)	A glycogen depletion phase: 65% V̇O_2max_ run in the heat (R1), 2 h recovery, 20 min running TT (R2)	Honey drink (HD) or plain water with an amount equivalent to 150% of body weight loss in 3 boluses (60%, 50%, and 40% subsequently) Honey dosage: 6.8% CHO	During the 2 h rehydration phase, subjects drank either plain water or honey drink (HD), equivalent to 150% of the body at 0, 30, and 60 min	↑HD vs. PL in R2
Vlahoyiannis et al. [81]	10 recreationally trained males	Morning CMJ test, a VRT and a 5 km cycling TT	HGI (109): jasmine and vegetables, supplying approximately 2 g·kg^−1^ CHO LGI (52): parboiled rice vegetables, supplying approximately 2 g·kg^−1^ CHO	Immediately post-exercise	↔HGI vs. LGI: 5 km TT, VRT and CMJ
Flynn et al. [72]	8 males (V̇O_2peak_: 56.7 ± 4.2 mL·kg^−1^·min^−1^) and 8 females (V̇O_2peak_: 46.5 ± 6.6 mL·kg^−1^·min^−1^)	90 min cycling glycogen depletion trial, rested for 4 h, 20 km cycling TT	SS: 1.6 g·kg^−1^ BM CHO sport supplement P: 1.6 g·kg^−1^ BM CHO potato-based food	0 and 2 h post-exercise (total 4 h recovery)	↔P vs. SS 20 km TT

CHO carbohydrate, VO_2max_ maximal oxygen consumption, V̇O_2peak_ peak oxygen consumption, CMJ countermovement jump, TT time-trial, VRT visual reaction test, Wmax Maximal power output, FF fast food, SS sport supplements.

## 5. Limitations and Future Directions

Scientific support for a food-first approach to CHO supplementation for endurance exercise performance is scarce, though some data do suggest that many foods are suitable CHO sources for endurance athletes. Nonetheless, strong conclusions are hindered by small samples sizes and a lack of appropriate controls, since many studies compared CHO intake in foods to water-only conditions. Many of the studies discussed here evaluated exercise performance using cycling protocols which, although suitable for cyclists, may not be applicable to runners or cross-country skiers who may be more affected by a full stomach should they aim to ingest recommended CHO doses via foods. Additionally, most studies recruited recreational or low-level athletes. Although these populations can still provide important information, results may not be directly applicable to top-level athletes due to training adaptations including CHO transport via training the gut [84]. More well-designed experimental studies with sufficient samples sizes to determine statistical significance and comparisons against “gold-standard” commercial CHO products such as bars, gels, drinks and powders are required to establish the extent to which these food sources can be considered optimal CHO for endurance exercise performance. This includes determining their efficacy as a pre-exercise, during exercise and post-exercise CHO source. It is particularly important that future lab-based studies investigating CHO provision throughout exercise provide these foods to adequately saturate different gut CHO transportable to accelerate higher CHO absorption and oxidation during training and competition. Further, future research is warranted to explore sex-based differences with regard to fuel utilization using different CHO foods before, during, and after endurance exercise. In addition, it is important to determine the extent to which these foods are tolerated by athletes across a wide range of different exercise modalities using validated questionnaires to determine feelings of stomach fullness and discomfort. Furthermore, the effectiveness of sustainably sourced CHO foods should be considered a priority for research in light of growing numbers of individuals in search of ethical alternatives.

## 6. Conclusions

A CHO-rich diet can optimise muscle glycogen stores and aid endurance exercise performance. While current recommendations for CHO ingestion commonly focus on energy and sports drinks, many athletes may wish to take a food-first approach where viable, although it must be noted that there are both pros and cons to this approach (Figure 4). Aside from CHO, many of these food sources provide protein, fats and fibres, vitamins and other micronutrients such as polyphenols that may also benefit endurance athletes. Direct research on many of the dietary CHO sources discussed here and endurance exercise is limited. All CHO sources could be considered interesting as compositions of meal plans throughout training and during the days leading up to competition to ensure muscle glycogen stores are maximized, although high-GI foods may be superior to low-GI foods to rapidly resynthesise muscle glycogen following exercise, and should thus be prioritised during intense competition. Although most of these CHO-rich foods appear equally effective for exercise performance compared to favoured CHO supplements such as drinks, gels and bars, not all of these food sources are equally viable during exercise due to difficulties in achieving the necessary quantities and ease of transport. Likewise, gastrointestinal discomfort appears more common with some of these food choices, likely due to the large quantities required to obtain recommended CHO doses. Endurance athletes can use the information provided herein to decide which CHO food source they may wish to trial before, during and/or following training and/or competition.

## Figures and Tables

**Figure 1 nutrients-15-01367-f001:**
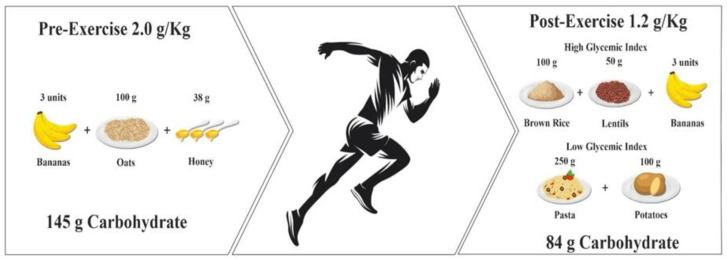
Examples of food-first options for pre- and post-exercise carbohydrate (CHO) provision for a 70 kg individual. Pre-exercise options can be from various CHO food sources with varying glycaemic index (GI), while post-exercise CHO provision can be high-GI.

**Figure 2 nutrients-15-01367-f002:**
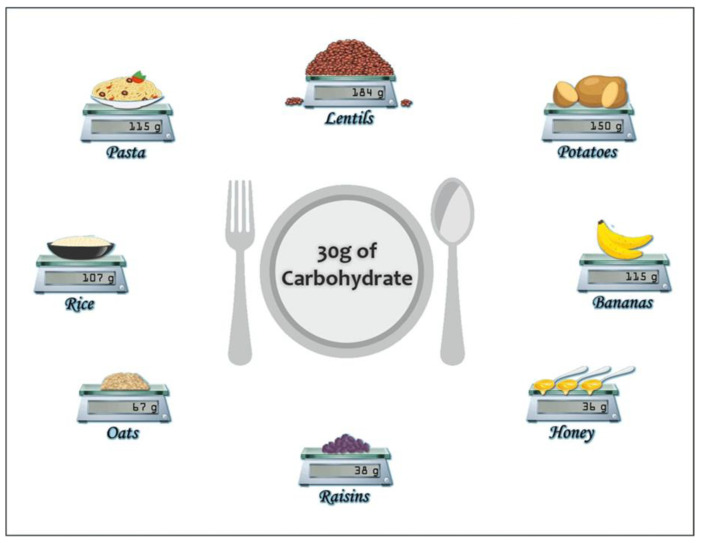
Amount (in g) of food sources that are required to achieve 30 g of carbohydrate (CHO). CHO dense foods such as honey, raisins and oats require far less total food than rice, bananas, pasta, potatoes or lentils to achieve 30 g of CHO.

**Figure 3 nutrients-15-01367-f003:**
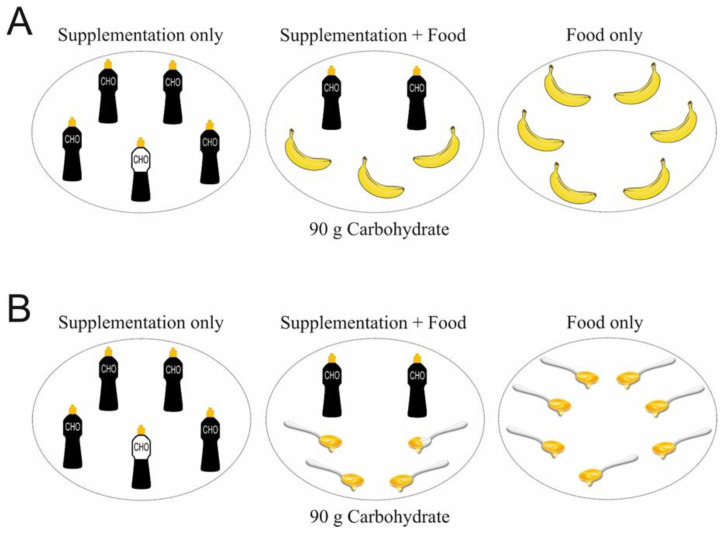
Examples of supplementation-only, food-only, and a combination of the two to provide 90 g·h^−1^ of carbohydrate (CHO) during exercise. Athletes can choose to replace some, or all, of their supplemental CHO with food choices. Panel (**A**): Bananas. Panel (**B**): Honey.

**Figure 4 nutrients-15-01367-f004:**
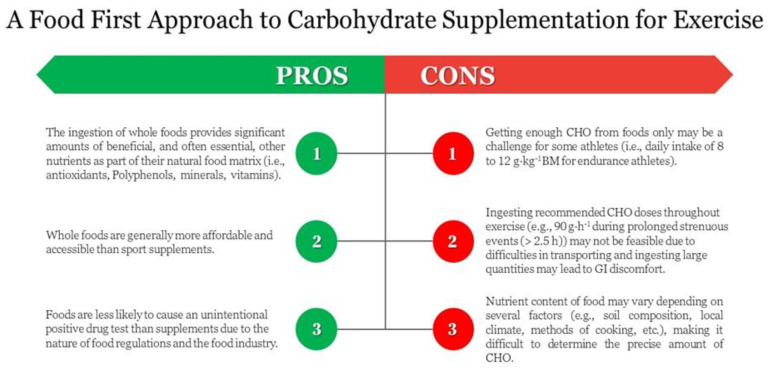
Some pros and cons of a food-first approach to carbohydrate supplementation for endurance exercise.

**Table 2 nutrients-15-01367-t002:** Summary of studies exploring the effects of CHO foods ingestion during exercise on performance.

Study	Subjects	Exercise Test	Treatments	Timing	Performance
Earnest et al. [52]	9 endurance-trained amateur males	64 km cycling TT on a cycle ergometer	Honey (H): (LGI = 35) Dextrose (D): (HGI = 100) PL: Artificially flavoured placebo	15 g of gel (honey, dextrose, or PL) with 250 mL water consumed every 16 km.	↑H vs. PL ↑D vs. PL ↔H vs. D
Rietschier et al. [53]	10 male endurance-trained cyclists and triathletes (V̇O_2max_: >45 mL·kg^−1^·min^−1^)	2 h cycling followed by a 10 km TT	1.1 g·kg^−1^ CHO from either six servings of 28 g raisin (R) or 26 g sports jellybeans (SJB)	Every 20 min during the 120 min cycling	↔ R vs. SJB
Too et al. [54]	11 healthy competitive male runners (V̇O_2max_: 58.2 ± 4.8 mL·kg^−1^·min^−1^)	80 min treadmill running at 75% V̇O_2max_ followed by a 5 km TT	Raisin (R): 0.7 g·kg^−1^ CHO Sport chew (SP): 0.7 g·kg^−1^ CHO	Pre-exercise: 0.5 g·kg^−1^ CHO and every 20 min during exercise: 0.2 g·kg^−1^ CHO	↑R vs. W ↑SP vs. W
Nieman et al. [49]	14 trained cyclists (V̇O_2max_: 58.7 ± 5.2 mL·kg^−1^·min^−1^)	75 km cycling TT	0.2 g·kg^−1^ CHO from bananas (B) or CHO beverage (CB)	Every 15 min throughout exercise.	↔B vs. CB
Nieman et al. [51]	20 male cyclists (V̇O_2max_: 51.0 ± 1.4 mL·kg^−1^·min^−1^)	75 km cycling TT	BAN: 0.4 g·kg^−1^ CHO from ripe Cavendish bananas PR: 0.4 g·kg^−1^ CHO from bosc pears W: Water only	20 min before exercise and 0.15 g·kg^−1^ CHO from BAN or PR every 15 min during exercise	↑BAN vs. W ↑PR vs. W
Nieman et al. [50]	20 cyclists 14 males (V̇O_2max_: 47.0 ± 1.5 mL·kg^−1^·min^−1^) 6 females (V̇O_2max_: 46.5 ± 2.8 mL·kg^−1^·min^−1^)	75 km cycling TT	CB: 0.4 g·kg^−1^ CHO from ripe Cavendish bananas MB: 0.4 g·kg^−1^ CHO from mini-yellow bananas SB: 6% Sugar beverage W: Water only	20 min before exercise and 0.2 g·kg^−1^ CHO from one of the two banana types or the 6% sugar beverage every 15 min during exercise	↔CB vs. MB vs. SB vs. W
Deli et al. [55]	11 healthy recreationally trained male (*n* = 9) and female (*n* = 2) adults (V̇O_2max_: 46.2 ± 1.9 mL·kg^−1^·min^−1^)	90 min of cycling at 60–70% V̇O_2max_, followed by a TT at 95% V̇O_2max_ to exhaustion	Corinthian currants (CC) GL: CHO glucose-drink W: Water only	1.5 g·kg ^−1^ CHO 30 min pre-exercise (7 mL·kg^−1^ BM) 3 mL·kg^−1^ BM every 20 min during the 90 min trial, and 7 mL·kg^−1^ BM within 15 min after exercise	↔CC vs. GL vs. W
Salvador et al. [56]	12 cyclists (V̇O_2peak_: 60.7 ± 9.0 mL·kg^−1^·min^−1^)	2 h cycle at 60–85% V̇O2_peak_ followed by a cycling TT (6 kJ·kg^−1^ BM)	P: 548 g of potato for 120 g of CHO Sports gels (SG): 184 g for 120 g CHO W: Water only	Servings to provide 15 g every 15 min during the test (128 g potato and 23 g SG per serving)	↑P vs. W ↑SG vs. W ↔P vs. SG

CHO carbohydrate, V̇O_2max_ maximal oxygen consumption, V̇O_2peak_ peak oxygen consumption, TT time-trial, TTE time to exhaustion, HGI high-glycaemic index, LGI low-glycaemic index meal.

## Data Availability

Not applicable.

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
