# Peer review of "Carbohydrates and Endurance Exercise: A Narrative Review of a Food First Approach"

_nutrients, 2023, doi:10.3390/nu15061367_

Round 1

Reviewer 1 Report

Thank you for the opportunity to review this work. The manuscript under this review discusses ‘food-first’ CHO sources for exercise performance with regards to pre, during and post-exercise consumption.

The study is intriguing, but due to its character as a review, it needs few improvements in order to be assessed as accepted for publication in Nutrients.

Abstract

The presentation of the abstract is clear.

I suggest the authors change the sentence in lines 25-26 at the end of the abstract, before the sentence in line 34 starting with "raising, banana..." for better understanding.

1. Introduction

Clearly explained and well complemented with all the data shown in the literature to date.

Line 70: what many more? Explain or change these words to "among others".

Lines 78-80: Rewrite this sentence, it is a bit confusing to understand.

2. Pre-exercise CHO ingestion

Line 90: Express the meaning of BM the first time it is presented in the manuscript.

Figure 1. Review the foods that you have marked as high and low GI, as there may have been an error in categorising them.

Line 155: The first time TT appears in the text, enter it, changing it to the one shown on line 164.

3. CHO ingestion during exercise

Well-presented information with the necessary considerations and the most relevant studies to provide relevant data from the narrative review.

4. Post-exercise CHO ingestion

The considerations presented for CHO reuptake are correct and well supported by the literature.

However, consider adding protein intake together with CHO in a 3:1 CHO:protein ratio to further aid rapid replenishment of glycogen stores in endurance sports.

5. Limitations and future directions

Well explained.

Do the authors consider that there might be differences in CHO supplementation between men and women in endurance sports?

6. Conclusions

Correct and defining the state presented by the authors in this narrative review.

Author Response

Reviewer 1

Authors’ responses

We would like to thank the reviewer for their constructive review of this manuscript. we hope this version satisfactorily addresses their points of concern. 

Reviewer’s comment

Thank you for the opportunity to review this work. The manuscript under this review discusses ‘food-first’ CHO sources for exercise performance with regards to pre, during and post-exercise consumption.

The study is intriguing, but due to its character as a review, it needs few improvements in order to be assessed as accepted for publication in Nutrients.

Authors’ responses

Thank you very much for your feedback and comments, we feel that the manuscript is now improved. We have responded to each suggestion in a point-by-point fashion.

Reviewer’s comment

Abstract

The presentation of the abstract is clear.

I suggest the authors change the sentence in lines 25-26 at the end of the abstract, before the sentence in line 34 starting with "raising, banana..." for better understanding.

Authors’ responses

Thanks, we have modified as suggested (Lines 25-26).

Reviewer’s comment

  1. Introduction

Clearly explained and well complemented with all the data shown in the literature to date.

Authors’ responses

Thanks.

Reviewer’s comment

Line 70: what many more? Explain or change these words to "among others".

Authors’ responses

We have revised the sentence for clarity (Lines 70-71).

Reviewer’s comment

Lines 78-80: Rewrite this sentence, it is a bit confusing to understand.

Authors’ responses

In contrast, high-GI CHO foods ingested during short-term recovery between exercise bouts may accelerate post-exercise muscle glycogen storage [21], and improve subsequent exercise performance [22]. Thus, different CHO-food sources with different GI may alter exercise performance and glycogen replenishment differently when recovery time is short.

Reviewer’s comment

  1. Pre-exercise CHO ingestion

Line 90: Express the meaning of BM the first time it is presented in the manuscript.

Authors’ responses

Done

Reviewer’s comment

Figure 1. Review the foods that you have marked as high and low GI, as there may have been an error in categorising them.

Authors’ responses

Our main goal for this figure is to show the exact doses that an endurance athlete needs to be consumed before and after exercise. GI ranking is not a goal of this figure.

Reviewer’s comment

Line 155: The first time TT appears in the text, enter it, changing it to the one shown on line 164.

Authors’ responses

Done

Reviewer’s comment

  1. CHO ingestion during exercise

Well-presented information with the necessary considerations and the most relevant studies to provide relevant data from the narrative review.

Authors’ responses

Many thanks to your great attention

Reviewer’s comment

  1. Post-exercise CHO ingestion

The considerations presented for CHO reuptake are correct and well supported by the literature.

However, consider adding protein intake together with CHO in a 3:1 CHO:protein ratio to further aid rapid replenishment of glycogen stores in endurance sports.

Authors’ responses

There is some evidence that protein may augment glycogen re-synthesis, which has been added into the manuscript. Line 446-449: “Further, it is important to highlight that CHO co-ingested with protein after exercise may enhance glycogen synthesis but only when the added energy of protein is ingested in addition to, not in replacement of, carbohydrates.”

Furthermore, the ratio of 3:1 CHO: protein has limited use and the current sport nutrition advice is to not use this ratio but rather focus on the absolute grams of carbohydrate and the relative dose of protein.

Reviewer’s comment

  1. Limitations and future directions

Well explained.

Do the authors consider that there might be differences in CHO supplementation between men and women in endurance sports?

Authors’ responses

Thanks for this question, we also added this:

Future research is warranted to explore sex differences with regards to fuel utilization using different carbohydrate foods to enhance endurance exercise.

Reviewer’s comment

  1. Conclusions

Correct and defining the state presented by the authors in this narrative review.

Authors’ responses

We appreciate it

Reviewer 2 Report

Carbohydrates and Endurance Exercise: A Narrative Review of a Food First Approach

This narrative review aims to discuss different “food-first” carbohydrate sources and their effects on athletic performance. Carbohydrate intake before, after and during exercise is covered.

The subject of the review is both timely and relevant and has been very well prepared. The narrative review covers all aspects of food-first carbohydrate sources and meets the nerve of the time because the awareness of natural food sources is increasing more and more. References are well chosen and discussed.

Below are just a few small comments to improve the review.

·        Line 116: The proportion of carbohydrates should also be given in g CHO per 100 g, as with the other sources.

·        Table 1-3: References should be ordered by year.

Author Response

Reviewer 2

Authors’ responses

We would like to thank the reviewer for their constructive review of this manuscript. we hope this version satisfactorily addresses their points of concern. 

Reviewer’s comment

Carbohydrates and Endurance Exercise: A Narrative Review of a Food First Approach

This narrative review aims to discuss different “food-first” carbohydrate sources and their effects on athletic performance. Carbohydrate intake before, after and during exercise is covered.

The subject of the review is both timely and relevant and has been very well prepared. The narrative review covers all aspects of food-first carbohydrate sources and meets the nerve of the time because the awareness of natural food sources is increasing more and more. References are well chosen and discussed.

Below are just a few small comments to improve the review.

  • Line 116: The proportion of carbohydrates should also be given in g CHO per 100 g, as with the other sources.

Authors’ responses

Both white and brown rice are predominantly composed of water (~69-70%) and contain an average of 28 and 26 g of CHO per 100 g, respectively. White or “refined” pasta is composed of ~67% water, and 100 g of plain cooked spaghetti provides ~26 g of CHO. Cooked lentils are another CHO food [30]….

Reviewer’s comment

 Table 1-3: References should be ordered by year.

Authors’ responses

We have revised as suggested.